# Role of Mineralocorticoid Receptor Antagonists in Diabetic Kidney Disease

**Maria-Eleni Alexandrou, Marieta P. Theodorakopoulou** and **Pantelis A. Sarafidis** *

Department of Nephrology, Hippokration Hospital, Aristotle University of Thessaloniki,
GR54642 Thessaloniki, Greece; alexandrou.me@gmail.com (M.-E.A.); marietatheod@gmail.com (M.P.T.)
* Correspondence: psarafidis11@yahoo.gr; Tel./Fax: +30-2313-312930

**Abstract:** Diabetic kidney disease (DKD) represents a major public health issue, currently posing an important burden on healthcare systems. Renin–angiotensin system (RAS) blockers are considered the cornerstone of treatment of albuminuric DKD. However, a high residual risk of progression to more advanced CKD stages under RAS blockade still remains, while relevant studies did not show significant declines in cardiovascular events with these agents in patients with DKD. Among several other pharmacological classes, mineralocorticoid receptor antagonists (MRAs) have received increasing interest, due to a growing body of high-quality evidence showing that spironolactone and eplerenone can significantly lower blood pressure and albuminuria in patients with CKD. Furthermore, finerenone, a novel nonsteroidal MRA with unique physicochemical properties, was shown to effectively reduce cardiovascular events and death, as well as the incidence of end-stage kidney disease in patients with type 2 diabetes. This review discusses previous and recent clinical evidence on the issue of nephroprotection and cardioprotection in DKD offered by mineralocorticoid receptor antagonism, aiming to aid clinicians in their treatment decisions for diabetic patients.

**Keywords:** diabetic kidney disease (DKD); mineralocorticoid receptor antagonists (MRAs); finerenone; spironolactone; eplerenone; esaxerenone; nephroprotection; albuminuria; end-stage kidney disease (ESKD); cardioprotection

## 1. Introduction

Diabetes represents a worldwide silent epidemic, affecting in 2019 9.3% of the adult population and accounting for 4 million deaths in 2017 [1]. The increase in diabetes prevalence has led to an increase in patients diagnosed with diabetic kidney disease (DKD), a complication occurring in 20–40% of diabetic patients [2]. DKD is the leading cause of end-stage kidney disease (ESKD) [3], with renal replacement therapy posing a major economic challenge in high-income countries (3–5% of national healthcare budgets in Europe) [4]. In patients with diabetes and chronic kidney disease (CKD), 10-year cumulative mortality is nearly three-fold higher compared to people with diabetes without CKD (31.1% vs. 11.5%) [4]. Thus, comprehension of the pathophysiology of DKD and identification of new directions for treatment strategies is crucial for health systems.

Over the last decades, the use of renin–angiotensin system (RAS) blockers has been the mainstay for retarding progression of DKD [5,6], along with lifestyle modifications and blood pressure (BP) and glycemic control [7]. Despite the indisputable nephroprotective effects of RAS blockers, accumulated evidence suggests persistence of a high residual risk for CKD and cardiovascular (CVD) progression in these patients, underlining the need for further research to establish novel treatment approaches. Double RAS blockade, initially associated with greater reductions in albuminuria compared to single blockade [8], was terminally abandoned in the form of a combination of an angiotensin converting enzyme inhibitor (ACEi) with an angiotensin receptor blocker [ARB], or of a renin inhibitor with an ACEi/ARB in diabetics and all other populations, in view of safety concerns raised

after publication of two major cardiovascular and renal outcome trials [9,10]. In contrast, sodium–glucose cotransporter-2 inhibitors (SGLT2i) have shown to offer kidney protection and are now included in recent KDIGO recommendations as preferred first line treatment in patients with type 2 diabetes mellitus (DM) and estimated glomerular-filtration rate (eGFR) $\geq$ 30 mL/min/1.73 m$^2$ [7].

Steroidal mineralocorticoid receptor antagonists (MRAs) have proven to be effective in the management of primary aldosteronism due to bilateral adrenal hyperplasia or aldosterone-producing adrenal adenomas [11], in the treatment of resistant hypertension [12,13], as well as in the reduction of albuminuria in patients with diabetic and non-diabetic nephropathy (alone or on top of an RAS blocker) [14]. Moreover, the accumulated clinical trial evidence supporting a significant reduction in mortality and CV events in patients with heart failure (HF) with reduced ejection fraction with MRAs changed the landscape of HF treatment [15–17], offering these drug agents a Class 1A recommendation [18]. Subgroup and post hoc analyses [15,19] of the latter studies showed improved clinical outcomes irrespective of baseline kidney function; however, patients with more advanced CKD stages (i.e., eGFR < 30 mL/min/1.73 m$^2$) had been excluded.

The risk of hyperkalemia remains a major determinant of MRA use, especially for the older first- and second-generation steroidal agents spironolactone and eplerenone [20]; an acute, usually reversible, decline in glomerular filtration observed among patients with low eGFR was another problem, particularly when these drugs were used in conjunction with an ACEi or an ARB [21]. In recent years, novel nonsteroidal agents have been developed, i.e., finerenone, esaxerenone and apararenone, with the former having been more extensively studied in diabetic populations. Finerenone belongs to third-generation MRAs and has demonstrated a more potent and selective antagonism for mineralocorticoid receptors (MR) compared to spironolactone and eplerenone, with a balanced distribution between heart and kidney tissue and different agonistic effects with regards to coregulatory molecules, rendering it a more potent anti-inflammatory and anti-fibrotic agent [22]. Large-scale clinical trials have recently provided evidence of improved renal and cardiovascular outcomes with addition of finerenone to the standard of care in patients with type 2 DM and moderately or severely increased albuminuria [23,24]. The aim of this review is to summarize available evidence on the effects of MRAs on BP, albuminuria and hard outcomes in patients with DKD, in whom background treatment has failed to delay CKD progression and novel drug regimens are being sought to address the residual CV risk.

## 2. Pleiotropic Effects of Aldosterone in Diabetic Kidney Disease

*2.1. Physiological Role of Aldosterone and Mechanisms of Ligand-Specific Activation of Mineralocorticoid Receptors*

Aldosterone is a steroid hormone that is produced by the adrenal cortex and acts, together with other members of the steroid hormone family (cortisol and corticosterone), as a ligand of MRs [25]. MRs are nuclear receptors, structurally similar to glucocorticoid receptors (GRs), that are expressed in epithelial and non-epithelial tissues, serving as transcription factors of target genes that regulate cellular processes [25] (Figure 1). In epithelial cells of the distal nephron, where both MR and GR are expressed, aldosterone exerts its classical actions with regards to volume depletion and hyperkalemia by regulating sodium, chloride and potassium handling, through transcription of the epithelial sodium channel (ENaC), $Cl^-/HCO_3^-$ exchangers and ROMK channels [26,27]. Activation of MR in non-epithelial tissues, including cardiomyocytes, smooth muscle cells, fibroblasts and macrophages in heart tissue [28,29], and glomerular podocytes [30], monocytes [31] and mesangial cells [32] in the kidney tissue, targets expression of genes that are involved in tissue repair and results in excess inflammation and fibrosis [22,33]. Activation of GRs modulates transcription of responsive genes related to energy homeostasis, response to stress and control of inflammation [25]. While only cortisol binds to the GR, the MR has multiple ligands with high affinity, including both cortisol and aldosterone [22,25]. Despite the fact that cortisol reaches up to 1000-fold higher concentrations in several tissues,

aldosterone is the primary physiological MR ligand in humans [34]. Pre-receptor conversion of cortisol by 11-beta-hydroxysteroid-dehydrogenase-2 (11βHSD2) to inactive cortisone is a key mechanism for maintaining MR selectivity in target tissues and regulating distinct MR functions in the heart and kidneys [33,35]. Differentiations in the concentrations of 11βHSD2 in each tissue represent the major modulator of this process, with the enzyme being abundantly expressed in distal tubular epithelial cells but not in cardiomyocytes, podocytes and macrophages, where cortisol is the primary physiological ligand of MRs [33].

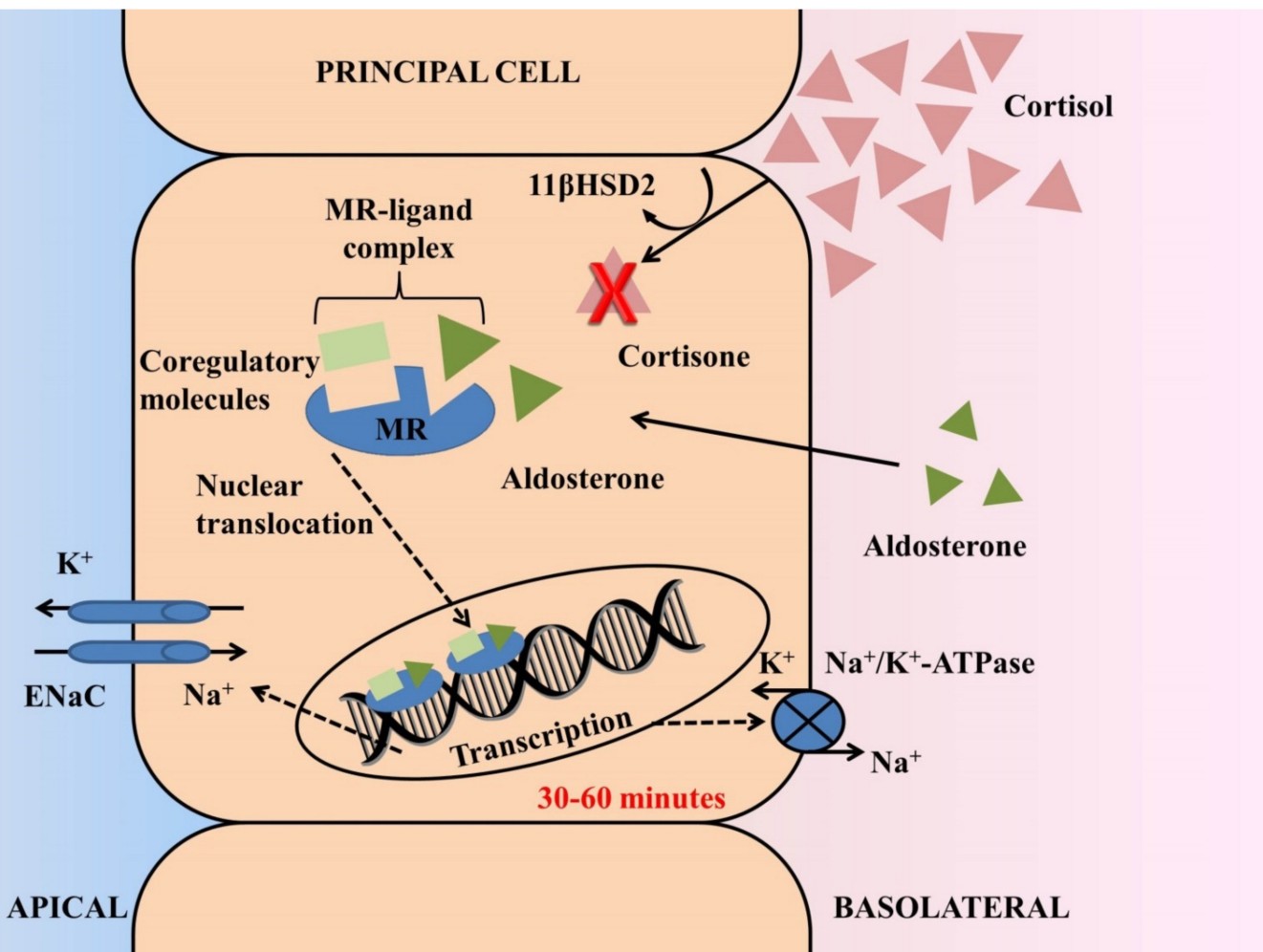

**Figure 1.** Actions of aldosterone through activation of the mineralocorticoid receptor (MR) in principal cells of the distal nephron. MR displays similar affinity for aldosterone and cortisol and ligand selectivity in the kidney is maintained by pre-receptor modulation. Despite the fact that cortisol reaches 100- to 1000-fold higher concentrations in several tissues, the 11-beta-hydroxysteroid-dehydrogenase-2 (11-HSD2) enzyme in tubular epithelial cells leads to conversion of cortisol to inactive cortisone, thereby rendering aldosterone the primary physiological MR ligand in the distal nephron. The genomic process begins after free diffusion of aldosterone through the cell membrane and fastening to the aldosterone ligand-binding domain of the MR. Upon ligand binding, transcription is dependent on recruitment of coregulatory proteins (coactivators or corepressors) and requires translocation of the MR-ligand complex into the nucleus. The final step of this process is upregulation of synthesis of the epithelial sodium channel (ENaC) at the apical membrane and of the Na/K–ATPase at the basolateral membrane, leading to renal sodium reabsorption and potassium secretion. In this classical genomic response, a latency of 30–60 min is expected after release or administration of aldosterone.

During the previous decade, the previously called nongenomic effects of aldosterone were elucidated. The genomic response is the process that includes all the classical steps of cell membrane diffusion of aldosterone: binding to the MR in the cytoplasm, translocation to the nucleus and activation of gene transcription [36,37]. This process results in an increase of ENaC concentration in epithelial cells (i.e., distal tubule, colon) and can be evidenced 30–60 min post aldosterone release/administration [38]. In addition to genomic actions, rapid effects of aldosterone have been described that cannot be explained by the traditional pathway, nor be blocked by inhibition of the transcriptional process molecules such as actinomycin D and aldosterone blockers, now considered to be mediated by MRs [39]. These rapid effects are associated with enhanced activity of the $Na^+-K^+-2Cl^-$ cotransporter and $Na^+-K^+$-ATPase in the heart, and of the $Na^+-H^+$ antiporter, the ENaC and $Na^+-K^+$-ATPase in the kidney, and are connected to subcellular trafficking [39].

In addition to the above, over the past years the crucial role of coregulatory molecules in mediating the genomic response to nuclear receptor activation has emerged [36]. Accumulated evidence suggests that upon ligand binding, the transcription of effector proteins is modulated via recruitment of coactivator or corepressor proteins according to distinct MR conformations induced by binding of different agonist ligands [25,36]. Ligand-selective peptides acting as potent antagonists of the MR-mediated transcription process were identified about a decade ago [36].

### 2.2. Aldosterone Breakthrough and Potential Mechanisms of Aldosterone-Induced Diabetic Kidney Disease and Cardiac Damage

In 10–53% of patients that initiate an ACEi/ARB, plasma aldosterone levels tend to rise again 6–12 months later, leaving them exposed to the deleterious proinflammatory and profibrotic effects of aldosterone in the kidneys, heart and vessels [26,40,41]. It has been speculated that this phenomenon of "aldosterone breakthrough" may represent a major cause of accelerated GFR decline in patients with type 1 diabetes and DKD and a poorer antiproteinuric response despite use of a single RAS blockade [33]. Inappropriate activation of MRs by aldosterone in podocytes, monocytes and mesangial cells in the kidney induces monocyte and macrophage infiltration [42] and promotion of glomerulosclerosis and interstitial fibrosis [43]. In the heart, overactivation of MRs promotes cardiac fibrosis, increased collagen synthesis and cardiac remodeling and hypertrophy [38,44,45]. An additional negative inotropic effect of aldosterone, counteracting the positive inotropic effect of angiotensin-II, has been also described [46].

Moreover, in the experimental model of streptozotocin-induced diabetes, a classical model of type 1 DM, overexpression of mRNA of MR, NADPH oxidase and collagen I/IV was described, resulting in collagen deposition in glomerular and tubulointerstitial areas in the kidney [47]. Local production of aldosterone in mesangial cells induced by angiotensin-II, high glucose and LDL has been also associated with the pathogenesis of DKD through MR activation [48]. Administration of spironolactone was shown to block MR overexpression [47] and reduce collagen deposition in streptozotocin-induced diabetic rats [49]. Similarly, administration of spironolactone, eplerenone and finerenone in classical experimental models of DKD due to type 2 DM led to amelioration of glomerulosclerosis and macrophage infiltration [50], prevention of podocyte injury [51] and reduction of proteinuria and NGAL expression [52].

The crucial role of GR in progression of DKD has recently emerged. In an experimental model of streptozotocin-induced diabetes in podocyte GR knockout mice, the loss of podocyte GR resulted in enhanced Wnt signaling, higher expression of transforming growth factor-β and β-catenin and disturbed fatty acid metabolism, accompanied by histological evidence of worsened fibrosis, increased collagen deposition, as well as mesenchymal transition of the glomerular endothelium and glomerulomegaly [53]. Similarly, the loss of the endothelial GR has been shown to induce upregulation of Wnt/β-catenin signaling and to promote angiogenesis and mesenchymal transition in tubular epithelial cells in diabetic experimental models [54,55]. The above findings suggest the presence of a podocyte–

endothelial cell crosstalk that is regulated by GR; this probably represents an additional mechanism in the development of diabetic nephropathy, on which more light should be shed in the future.

## 3. Clinical Studies of Mineralocorticoid Receptor Antagonism in Diabetic Kidney Disease

After considerable preclinical data yielding information on the pivotal role of MR overactivation in the progression of kidney and cardiac injury were available, several clinical studies embarked to investigate the potential benefits from MR blockade in patients with diabetic and non-diabetic kidney disease.

During the previous two decades, the non-selective MRA spironolactone and the selective eplerenone were extensively studied mainly for their anti-albuminuric effects. These older classical steroidal MRAs function as active competitors to aldosterone for binding to the MR ligand-binding pocket, destabilizing the active conformation of the receptor [56]. They prevent recruitment of transcriptional coactivators in the presence of aldosterone, but in the absence of aldosterone they exhibit partially agonistic coactivator recruitment properties [22]. They both lack tissue and ligand specificity, while spironolactone additionally lacks receptor specificity [56]. Canrenone, an active metabolite of spironolactone, was expected to have fewer side effects than spironolactone by preventing the formation of intermediate products with antiandrogenic and progestational actions [57]. It was approved for clinical use in Europe, however, its use was limited by significant hyperkalemia associated with lack of receptor-specific selectivity, similar to a first-generation MRA [58].

Finerenone is a third-generation nonsteroidal MRA that in vitro inhibits binding of coregulatory molecules independent of the presence or absence of aldosterone; it is characterized by tissue and ligand specificity and equal distribution between heart and kidney [26]. Equinatriuretic to eplerenone doses of finerenone demonstrate stronger anti-inflammatory and anti-fibrotic effects [59]. Esaxerenone is an equally potent and selective nonsteroidal MRA also exhibiting ligand-specific binding properties [60] and a dose-dependent BP-lowering effect that is at least equivalent to eplerenone, according to the Esaxerenone and Eplerenone in Patients With Essential Hypertension (ESAX-HTN Study) [61]. Apararenone is another compound belonging in the nonsteroidal family, also exhibiting highly selective MRA activity [62].

### 3.1. Effect of Mineralocorticoid Receptor Antagonists in Albuminuria/Proteinuria and Kidney-Related Outcomes

3.1.1. Spironolactone

In 2003, Sato et al. [63] were the first to study the clinical effect of aldosterone breakthrough in diabetic populations (Table 1). They followed 45 patients with type 2 DM, UACR 30–300 mg/g and creatinine clearance >60 mL/min for 40 weeks and observed a rise in aldosterone levels in 40% of them, despite treatment with trandolapril. In 15 of these 18 patients, the antiproteinuric effect of ACEi had declined to baseline; among them 13 patients were assigned to additionally receive spironolactone 25 mg for 24 weeks, resulting in a significant decrease in albuminuria and left ventricular mass index. In 2006, Chrysostomou et al. [64] evaluated 41 patients with overt proteinuria > 1.5 g/day and showed that the addition of spironolactone alone to a background treatment with ramipril produced a higher proteinuria reduction than addition of irbesartan and an equal reduction to the addition of both spironolactone and irbesartan.

**Table 1.** Main studies on the effects of spironolactone in albuminuria, renal and cardiovascular outcomes in patients with DKD.

| Study | Study Design | Follow Up | Nr of Patients | Patient Characteristics | Comparator | Back-Ground Treatment | Main Results |
|---|---|---|---|---|---|---|---|
| Sato et al., 2003 [63] | Open-label, single-arm study | 24 weeks | 13 | T2DM, UACR 30–300 mg/g and creatinine clearance > 60 mL/min | - | Trandolapril | ↓ UAE and LVMI for spironolactone ($p < 0.05$ for both) Non-significant ↑ K (before $4.2 \pm 0.3$ mEq/L; after $4.3 \pm 0.2$ mEq/L) |
| Rossing et al., 2005 [65] | Double-blind, cross-over RCT | 8 weeks | 21 | T2DM and UAE > 300 mg/24 h | Placebo | ACEi or ARB | ↓ UAE −33% (95%CI −41%, −25%); 24 h SBP −6 mmHg (95%CI −10, −2); 24 h DBP −4 mmHg (95%CI −6, −2); eGFR −3 mL/min/1.73 m$^2$ (95%CI −0.3, +6.0) for spironolactone |
| Schjoedt et al., 2005 [66] | Double-blind, cross-over RCT | 8 weeks | 20 | T1DM and UAE > 300 mg/24 h | Placebo | ACEi or ARB | ↓ UAE −30% (95%CI −41%, −17%); 24 h SBP −8 mmHg (95%CI −17, +1); 24 h DBP −3 mmHg (95%CI −7, +0.2); eGFR by −3.4 mL/min/ 1.73 m$^2$ (−6.9, 0.1) for spironolactone 1 patient excluded due to hyperK in spironolactone |
| Schjoedt et al., 2006 [67] | Double-blind, cross-over RCT | 8 weeks | 20 | T1DM or T2DM and UAE > 2500 mg/24 h | Placebo | ACEi or ARB | ↓ UAE −32% (95%CI −42%, −21%); 24 h SBP −6 mmHg (95%CI −10, −2); 24 h DBP −4 mmHg (95%CI −6, −2); eGFR −3 mL/min/1.73 m$^2$ (−6.0, +1.0) ↑ K +0.2 mmol/L (95%CI −0.004, +0.5) |
| van den Meiracker, 2006 [68] | Double-blind, parallel-group RCT | 1 year | 59 | T2DM and UAE > 300 mg/24 h or UACR > 20 mg/g | Placebo | ACEi or ARB | ↓ UAE −40.6% (95%CI −57.8%, −23.4%); 24 h SBP −7 mmHg (95%CI −12, −2); 24 h DBP −3 mmHg (95%CI −6, −1) eGFR −12.9 mL/min/1.73 m$^2$ (−16.5, −9.5) for spironolactone; −4.9 mL/min/1.73 m$^2$ (−8.9, −0.8) for placebo 5 patients excluded due to hyperK in spironolactone |
| Saklayen et al., 2008 [69] | Double-blind, cross-over RCT | 7 weeks | 30 | T1DM or T2DM patients with any level of proteinuria | Placebo | ACEi or ARB | ↓ UPCR from $1.80 \pm 1.78$ to $0.79 \pm 0.99$ for spironolactone ($p = 0.004$); from $1.24 \pm 1.13$ to $1.57 \pm 2.13$ for placebo ($p = 0.35$); eGFR from $61.91 \pm 23.4$ to $53.94 \pm 23.58$ for spironolactone ($p = 0.0001$) |
| Mehdi et al., 2009 [70] | Double-blind parallel-group RCT | 48 weeks | 81 | T1DM or T2DM patients and UACR > 300 mg/g | Placebo, Losartan 100 mg | Lisinopril 80 mg | ↓ UACR −34% (95%CI −51%, −11.2%) for spironolactone ($p = 0.007$ vs. placebo); −16.8% (95%CI −37.3%, +10.5%) for losartan ($p = 0.2$ vs. placebo); % change in creatinine clearance −13.1% (95%CI −21.3%, −3.9%) for spironolactone; −16.8% (95%CI −23.9%, −9.1%) for losartan; −16.0% (95% CI −23.3%, −7.9%) for placebo HyperK episodes (>6.0 mmol/L): 14 patients in spironolactone ($p < 0.001$ vs. placebo); 10 in losartan ($p = 0.009$ vs. placebo); 2 patients in placebo |

**Table 1.** *Cont.*

| Study | Study Design | Follow Up | Nr of Patients | Patient Characteristics | Comparator | Back-Ground Treatment | Main Results |
|---|---|---|---|---|---|---|---|
| Nielsen et al., 2012 [71] | Double-blind, cross-over RCT | 60 days | 21 | T1D and UAE > 30 mg/day | Placebo | ACEi or ARB | ↓ UACR −60% (range −80% to −21%); eGFR from 78 ± 6 to 72 ± 6 mL/min/1.73 m$^2$ ($p = 0.003$) HyperK episodes (>5.7 mmol/L): 2 patients in spironolactone group |
| Ziaee et al., 2013 [72] | Parallel-group RCT | 12 weeks | 60 | T2DM and microalbuminuria | Placebo | Enalapril | ↓ UACR from 126 ± 69.3 to 59.3 ± 48.1 for spironolactone ($p < 0.001$); eGFR from 79.8 ± 18 to 75.6 ± 16.3 mL/min/1.73 m$^2$ for spironolactone ($p = 0.6$) |
| Esteghamati et al., 2013 [73] | Open-label, parallel-group RCT | 18 months | 136 | T2DM and UAE ≥ 30 mg/day | Enalapril | Losartan | ↓ UAE −60.5 mg (95%CI −148.8, −16.4) for spironolactone; +22.0 mg (95%CI −110.3, +108.9) for placebo ($p = 0.017$); SBP −8.89 mmHg (95%CI −15.88, −1.89) for spironolactone; −6.08 mmHg (−14.71, +2.57) for placebo ($p < 0.001$); DBP −4.44 mmHg (95%CI −8.10, −0.79) for spironolactone; −2.86 (−7.06, +1.34) for placebo ($p = 0.001$); eGFR −10.23 mL/min/1.73 m$^2$ (95%CI −16.69, −3.76) for spironolactone; −9.08 mL/min/1.73 m$^2$ (−16.06, −2.10) for placebo ($p = 0.674$) |
| Oxlund et al., 2013 [13] | Double-blind, parallel-group RCT | 16 weeks | 119 | T2DM and resistant hypertension | Placebo | ACEi or ARB | ↓ UACR −7.3 mg/g (95%CI −1093, +12.2) for spironolactone; +0 mg/g (95% +7, +146.3) for placebo ($p = 0.001$); placebo-corrected 24 h SBP −8.9 mmHg (95%CI −13.2, −4.6); placebo-corrected 24 h DBP −3.9 mmHg (95%CI −6.2, −1.7) ↑ K +0.26 mmol/L (95%CI +0.1, +0.4) for spironolactone; +0.02 (95%CI +0.07, +0.10) for placebo (between-group $p < 0.001$) |
| Kato et al., 2015 [74] | Open-label parallel-group RCT | 8 weeks | 52 | T2DM and UACR 100–2000 mg/g | Placebo | ACEi or ARB | ↓ UACR −33% (95%CI −54%, −22%); eGFR −3.2 ± 9.7 mL/min/1.73 m$^2$ ($p = 0.052$) |
| Chen et al., 2018 [75] | Open-label, parallel-group RCT | 72 weeks | 244 | T2DM and UAER 20–199 μg/min | Placebo | Irbesartan 150 mg or 300 mg | ↓ UAER −30 μg/min (95%CI −54, −15) for spironolactone/irbesartan 300 mg; −30 μg/min (95%CI −51, −12) for spironolactone/irbesartan 150 mg; −23 μg/min (95%CI −35, −12) for irbesartan 300 mg; −15 μg/min (95%CI −24, −11) for irbesartan 150 mg (between-group $p < 0.001$) |

↓, decrease; ↑, increase; ACEi, angiotensin converting enzyme inhibitor; ARB, angiotensin receptor blocker; CI, confidence interval; DBP, diastolic blood pressure; eGFR, estimated glomerular filtration rate; hyperK, hyperkalemia; K, potassium; LVMi, left ventricular mass index; RCT, randomized controlled trial; SBP, systolic blood pressure; T2DM, type 2 diabetes mellitus; UACR, urinary albumin creatinine ratio; UAER, urinary albumin excretion rate.

Subsequently, Rossing et al. randomly assigned 21 type 2 DM patients with urinary albumin excretion (UAE) > 300 mg/24 h to receive spironolactone 25 mg or placebo added to a maximally up-titrated ACEi or ARB for 8 weeks in a cross-over design study [65]. The authors observed a significant reduction in albuminuria by −33% for spironolactone, as well as higher reductions in ambulatory 24 h SBP and DBP (−6 mmHg and −4 mmHg) compared to placebo ($p < 0.001$ for both comparisons). A non-significant and reversible eGFR decline by −3 mL/min/1.73 m$^2$ was also observed. In subsequent trials from the same center, similar effects on UAE, 24 h BP and eGFR were reported with addition of spironolactone in 20 type 1 DM patients with UAE > 300 mg/24 h [66], as well as in 20 type 1 and type 2 DM patients with UAE > 2500 mg/24 h [67]. In a study by van der Meirecker et al. [68], 59 type 2 DM patients with UAE > 300 mg/day or UACR > 20 mg/g were evaluated for 1 year, to examine effects of spironolactone (25–50 mg) compared to placebo on top of an ACEi/ARB on proteinuria, BP and renal function. A significant decrease in albuminuria by −41% and in BP by −7/−3 mmHg was noted with spironolactone but not with placebo. In the spironolactone group, the reduction in eGFR was largest during the first 3 months after initiation of treatment but subsided during further follow-up. The slope of eGFR decline was similar between the spironolactone and placebo groups 3 months after allocation to study medication.

In perhaps one of the most detailed studies with UACR as the primary endpoint, Mehdi et al. [70] randomly allocated 81 patients with type 1 or type 2 DM and UACR > 300 mg/g, already under maximum dose of lisinopril (80 mg/day) to receive placebo, losartan (100 mg daily) or spironolactone (25 mg daily) for 48 weeks. They observed a decrease in UACR by −34.0% for spironolactone and by −16.8% for losartan. No significant between-group differences were noted in changes of creatinine clearance (−13.1% for spironolactone, −16.8% for losartan and −16% for placebo, $p = 0.8$). Several subsequent studies confirmed the additional anti-albuminuric effect of spironolactone when administered on top of an ACEi/ARB in diabetic patients with moderately or severely increased albuminuria [69,71,72,74–76] (Table 1).

### 3.1.2. Eplerenone

In a large study by Epstein et al. [77] in 2006, 268 patients with type 2 DM and UACR ≥ 50 mg/g were randomly assigned to receive placebo, eplerenone 50 mg or eplerenone 100 mg on top of enalapril 20 mg for 12 weeks (Table 2). A significant reduction of UACR by −41% and −48.4% was noted in the two eplerenone groups, respectively, but only by −7.4% in the placebo group ($p < 0.001$ vs. placebo for both comparisons). Systolic and diastolic BP decreased significantly in all three treatment groups, without between-group significant differences being observed. Overall, these results suggested that the anti-albuminuric effect of MR blockade was independent of BP reduction. The incidence of sustained (>5.5 mmol/L on two consecutive measurements) and severe (≥6.0 mmol/L at any timepoint) hyperkalemia did not significantly differ among the three treatment arms. A significant reduction in UACR by −70% was shown with an eplerenone/ramipril combination compared to ramipril monotherapy (−37%) in a recent smaller single-blind study in 75 type 2 DM patients with UACR 30–300 mg/g ($p < 0.0001$) [78]. In patients with eGFR < 60 mL/min/1.73 m$^2$, a higher incidence of sustained hyperkalemia (>5.5 mmol/L on 2 consecutive occasions) was noted in the eplerenone/ramipril combination group.

**Table 2.** Main studies on the effects of eplerenone in albuminuria, renal and cardiovascular outcomes in patients with DKD.

| Study | Study Design | Follow Up | Nr of Patients | Patient Characteristics | Comparator | Back-Ground Treatment | Main Results |
|---|---|---|---|---|---|---|---|
| Epstein et al., 2006 [77] | Double-blind, parallel-group RCT | 12 weeks | 268 | T2DM and UACR ≥ 50 mg/g | Placebo | Enalapril 20 mg | ↓ UACR −41% for eplerenone 50 mg; −48.4% for eplerenone 100 mg; −7.4% for placebo ($p < 0.001$ vs. placebo for both) Between-group differences in sustained hyperK (>5.5 mmol/L on two consecutive measurements) $p = 0.29$; severe hyperK (≥6.0 mmol/L at any timepoint) $p = 0.38$ |
| Brandt-Jacobsen et al., 2020 (MIRAD trial) [79] | Double-blind, parallel-group RCT | 26 weeks | 140 | T2DM, median UACR 17 mg/g, 12% had eGFR < 60 mL/min/1.73 m$^2$ | Placebo | Antihypertensive treatment | ↓ UACR by −34% for eplerenone vs. placebo ($p = 0.005$); eGFR −3.5 mL/min/1.73 m$^2$ for eplerenone Between-group differences in episodes of hyperkalemia (≥5.5 mmol/L) $p = 0.276$ ↑ K by +0.26 mmol/L for eplerenone |
| Mokadem et al., 2020 [78] | Single-blind, parallel-group RCT | 24 weeks | 75 | T2DM and UACR 30–300 mg/g and stage 1 hypertension | Treatment groups: Eplerenone/ramipril combination, ramipril monotherapy, eplerenone monotherapy | | ↓ UACR −70% for eplerenone/ramipril; −37% for ramipril; −38% for eplerenone ($p < 0.0001$ for combination vs. both others) HyperK episodes (>5.5 mmol/L on 2 measurements): 8% for eplerenone/ramipril; 4% for ramipril; 4% for eplerenone (for eplerenone/ramipril vs. others $p = 0.5$, ramipril vs. eplerenone $p = 0.6$); for eGFR < 60 mL/min/1.73 m$^2$: ↑ incidence of hyperK for eplerenone/ramipril vs. others ($p < 0.05$) |

↓, decrease; ↑, increase; eGFR, estimated glomerular filtration rate; hyperK, hyperkalemia; K, potassium; RCT, randomized controlled trial; T2DM, type 2 diabetes mellitus; UACR, urinary albumin creatinine ratio.

In a pre-specified secondary analysis of the Mineralocorticoid Receptor Antagonist in Type 2 Diabetes (MIRAD) trial by Brandt-Jacobsen et al. [79], 140 patients with type 2 DM and high risk of CVD (baseline UACR 17 mg/g, 12% of patients had an eGFR < 60 mL/min/1.73 m$^2$), that were randomly allocated to either a high dose of eplerenone (100–200 mg) or placebo on top of background antihypertensive treatment for 26 weeks, were evaluated for UACR reduction, incidence of hyperkalemia and kidney-related adverse events. A decrease of UACR by −34% was observed for eplerenone compared to placebo ($p = 0.005$), an effect that was consistent across subgroups defined by the presence of UACR ≥ 30 mg/g (p-interaction = 0.441) and treatment with a RAS blocker. No significant differences were noted in the incidence of mild hyperkalemia (≥5.5 mmol/L) ($p = 0.276$); initiation of eplerenone, however, was associated with an increase in serum potassium by 0.26 mmol/L and a decrease in eGFR of −3.5 mL/min/1.73 m$^2$.

### 3.1.3. Finerenone

The Mineralocorticoid Receptor Antagonist Tolerability Study–Diabetic Nephropathy (ARTS-DN) Study, published in 2015 by Bakris et al. [80], was the first placebo-controlled randomized clinical trial to investigate the safety and efficacy of different oral doses of finerenone compared to placebo, in 821 patients with type 2 DM and high or very high albuminuria on top of an RAS blocker (Table 3). After 90 days of treatment, dose-dependent reductions in placebo-corrected mean ratios of UACR relative to baseline were evidenced for finerenone (0.79, 0.76, 0.67 and 0.62 for the finerenone 7.5, 10, 15, 20 mg/day dose groups respectively); the risk of hyperkalemia leading to study discontinuation was 2.1%, 0%, 3.2% and 1.7%, respectively. In ARTS-DN Japan [81], in a subsequent similar study in 96 patients with type 2 DM and DKD, Katayama et al. [54] reported a numerical reduction of UACR at day 90 relative to baseline for each finerenone treatment group compared with placebo. Small mean increases were observed in serum potassium; no patient experienced an increase to levels ≥ 5.6 mmol/L or > 6 mmol/L throughout the study.

**Table 3.** Main studies on the effects of finerenone in albuminuria, renal and cardiovascular outcomes in patients with DKD.

| Study | Study Design | Follow UP | Nr of Patients | Patient Characteristics | Comparator | Back-Ground Treatment | Main Results |
|---|---|---|---|---|---|---|---|
| Bakris et al., 2015 (ARTS-DN) [80] | Double-blind, parallel-group RCT | 90 days | 821 (4 different finerenone dose groups) | T2DM and UACR 30 to <300 mg/g or >300 mg/g (stratified randomization) | Placebo | ACEi or ARB | ↓ placebo-corrected mean ratio of UACR at day 90 relative to baseline: finerenone 7.5 mg 0.79 ($p = 0.004$); finerenone 10 mg 0.76 ($p = 0.001$); finerenone 15 mg 0.67 ($p < 0.001$); finerenone 20 mg 0.62 ($p < 0.001$). Significantly ↑ incidence of hyperK episodes leading to study discontinuation: finerenone 7.5 mg 2.1%, finerenone 15 mg 3.2% and finerenone 20 mg 1.7%. No significant ↑ in the risk of hyperkalemia for placebo and finerenone 10 mg |
| Katayama et al., 2017 [81] | Double-blind, parallel-group RCT | 90 days | 96 (4 different finerenone dose groups) | T2DM and UACR 30 to <300 mg/g or >300 mg/g (stratified randomization) | Placebo | ACEi or ARB | ↓ LS mean ratio of finerenone to baseline (0.712); LS mean ratio of finerenone to placebo (0.670) for finerenone 20 mg ($p = 0.0240$). ↑ K for finerenone (+0.025, +0.167 mmol/L) vs. placebo (−0.075 mmol/L) |
| Bakris et al., 2020 (FIDELIO-DKD) [23] | Double-blind, parallel-group RCT | 2.6 years | 5734 | T2DM and: (a) UACR 300–5000 mg/g and eGFR 25–75 mL/min/1.73 m$^2$ or (b) UACR 30–300 mg/g, eGFR 25–60 mL/min/1.73 m$^2$, diabetic retinopathy | Placebo | ACEi or ARB | Primary composite endpoint of kidney failure (ESKD or eGFR < 15 mL/min/1.73 m$^2$), eGFR decrease of ≥40%, renal death: HR 0.82; 95% CI 0.73–0.93. Secondary: kidney failure HR 0.87; 95% CI 0.72–1.05; eGFR decrease of ≥40% HR 0.81; 95% CI 0.72–0.92. Secondary composite endpoint of kidney failure (ESKD or eGFR <15 mL/min/1.73 m$^2$), eGFR decrease of ≥57%, renal death: HR 0.76; 95% CI 0.65–0.90. Secondary composite endpoint of CV death, nonfatal MI/stroke, HHF: HR 0.86; 95% CI 0.75–0.99. Secondary: CV death HR 0.86; 95% 0.68–1.08; nonfatal MI HR 0.80; 95% CI 0.58–1.09; nonfatal stroke: HR 1.03; 95%CI 0.76–1.38; HHF HR 0.86; 95%CI 0.68–1.08 HyperK leading to drug discontinuation: 2.3% for finerenone; 0.9% for placebo. Pre-specified secondary analysis: new-onset AF HR 0.71, 95%CI 0.53–0.94; fatal/nonfatal stroke after new-onset AF: HR 7.13; 95%CI 4.01–12.70 |
| Pitt et al., 2021 (FIGARO-DKD) [24] | Double-blind, parallel-group RCT | 3.4 years | 7437 | T2DM and: (a) UACR 30–300 mg/g, eGFR ≥ 25–90 mL/min/ 1.73 m$^2$ or (b) UACR 300–5000 mg/g, eGFR ≥ 60 mL/min/1.73 m$^2$ | Placebo | ACEi or ARB | Primary composite endpoint of CV death, nonfatal MI/stroke, HHF: HR 0.87; 95% CI 0.76–0.98. Secondary: HHF HR 0.71; 95%CI 0.56–0.90. Secondary composite endpoint of kidney failure (ESKD or eGFR <15 mL/min/1.73 m$^2$), eGFR decrease of ≥40%, renal death: HR 0.87; 95%CI 0.76–1.01. Secondary: ESKD HR 0.64, 95%CI 0.41–0.995 HyperK leading to drug discontinuation: 1.2% finerenone; 0.4% placebo |

↓, decrease; ↑, increase; ACEi, angiotensin converting enzyme inhibitor; ARB, angiotensin receptor blocker; CI, confidence interval; CV, cardiovascular; eGFR, estimated glomerular filtration rate; ESKD, end-stage kidney disease; HHF, hospitalization for heart failure; hyperK, hyperkalemia; K, potassium; HR, hazard ratio; LS, least-squares; RCT, randomized controlled trial; T2DM, type 2 diabetes mellitus; UACR, urinary albumin creatinine ratio.

The clinical trials discussed above were focused on changes of albuminuria and were often not adequately powered to detect changes in the slope of eGFR or progression of CKD and CVD. Despite the fact that changes in albuminuria have proven to be strongly associated with a lower hazard for renal clinical endpoints (ESKD, fall of eGFR < 15 mL/min/1.73 m$^2$) and lower risk of death [82,83], these represent an intermediate marker of kidney disease. Thus, from an efficacy point of view, the only currently available data on the impact of MRAs on long-term cardiorenal protection originate from the two recently published large-scale phase 3 randomized clinical trials of finerenone, the FIDELIO-DKD (Finerenone in reducing kidney failure and disease progression in Diabetic Kidney Disease) [23] and the FIGARO-DKD (Finerenone in reducing cardiovascular mortality and morbidity in Diabetic Kidney Disease) [24].

In the FIDELIO-DKD study, 5734 patients with type 2 DM and (a) UACR 300–5000 mg/g and eGFR 25 to < 75 mL/min/1.73 m$^2$ or (b) UACR 30 to < 300 mg/g, eGFR 25 to < 60 mL/min/1.73 m$^2$ and diabetic retinopathy, were randomly assigned to finerenone 10–20 mg or placebo on top of a maximum tolerated dose of RAS blocker [23]. During a median of 2.6 years, the primary outcome event, a composite of kidney failure (ESKD or eGFR < 15 mL/min/1.73 m$^2$), sustained (≥4 weeks) eGFR decrease of ≥40% from baseline or death from renal causes, occurred in 17.8% of patients in the finerenone and 21.1% of patients in the placebo group (HR 0.82; 95%CI 0.73–0.93). Notably, finerenone was associated with an even larger reduction in the hazard of the main secondary renal outcome, a composite of kidney failure, sustained eGFR decrease of ≥57% from baseline or renal death (HR 0.76; 95% CI 0.65–0.90). Mean increases of serum potassium of about 0.23 mmol/L were observed with finerenone, while slightly higher rates of hyperkalemia episodes leading to drug discontinuation were reported with finerenone compared to placebo (2.3% vs. 0.9%). In addition to the above, lower rates of the key secondary cardiovascular outcome, a composite of time to first occurrence of cardiovascular death, nonfatal MI, nonfatal stroke or hospitalization for heart failure (HHF), were observed for patients receiving finerenone (13.0%) than in those receiving placebo (14.8%) (HR 0.86; 95%CI 0.75–0.99) [10].

In the FIGARO-DKD study, 7437 patients with type 2 DM and (a) UACR ≥ 30 to < 300 mg/g and eGFR ≥ 25–90 mL/min/1.73 m$^2$ or (b) UACR ≥ 300 to ≤ 5000 mg/g and eGFR ≥ 60 mL/min/1.73 m$^2$, were randomly assigned to receive finerenone 10–20 mg or placebo on top of a maximum tolerated dose of RAS blocker [24]. During a median follow-up of 3.4 years, the primary cardiovascular outcome event, a composite of cardiovascular death, nonfatal MI, nonfatal stroke or HHF, occurred in 12.4% of patients in the finerenone and 14.2% of patients in the placebo group (HR 0.87; 95%CI 0.76–0.98). Of importance, among the individual components of the primary outcome the greatest benefit was observed in reduction of risk for HHF (HR 0.71; 95%CI 0.56–0.90). Finerenone was also associated with a marginally but not statistically significant lower risk for the key secondary renal outcome, a composite of kidney failure, sustained eGFR decrease of ≥40% from baseline or renal death (HR 0.87; 95%CI 0.76–1.01). With regards to different components of the secondary outcome, significantly lower rates of progression to ESKD were observed with finerenone (0.9%) than with placebo (1.3%) (HR 0.64; 95%CI 0.41–0.995). The frequency of hyperkalemia episodes leading to drug discontinuation was even lower than in the FIDELIO-DKD study (1.2% vs. 0.4%, respectively).

The FIDELITY analysis, published at the end of 2021, was a pre-specified pooled efficacy and safety analysis of both the FIDELIO-DKD and FIGARO-DKD studies that together enrolled 13,171 patients, with a mean eGFR of 57.6 mL/min/1.73 m$^2$ and a UACR of 515 mg/g [84]. According to the full analysis set, finerenone was associated with a significant reduction of 14% in the risk for the composite cardiovascular outcome of time to cardiovascular death, nonfatal MI, nonfatal stroke or HHF (HR 0.86; 95%CI 0.78–0.95), and a significant reduction of 23% in the risk for the composite kidney outcome of time to first onset of kidney failure, sustained eGFR decrease ≥ 57% from baseline or renal death (HR 0.77; 95%CI 0.67–0.88). The cardiovascular benefit was primarily driven by a lower incidence of HHF (HR 0.78; 95%CI 0.66–0.92). The renal benefit was attributed to a 30%

reduction in the risk of sustained eGFR decrease $\geq 57\%$ (HR 0.70; 95%CI 0.60–0.83), as well as to a 20% reduction in the risk for ESKD (HR 0.80; 95%CI 0.64–0.99). It should be further emphasized that these results represent the first available evidence on beneficial effects of an MRA agent on hard renal and cardiovascular outcomes. Hyperkalemia leading to drug discontinuation was more frequently reported in patients receiving finerenone (1.7%) than in placebo (0.6%). No fatal hyperkalemia-related adverse event was reported.

The cardiovascular benefit obtained by finerenone administration in patients with DKD seems to extend beyond decreases in HHF. According to a secondary analysis of the FIDELIO-DKD study by Filippatos et al. [85], a significant reduction of 29% was evidenced in the risk of new-onset atrial fibrillation (AF) (HR 0.71; 95%CI 0.53–0.94), that was partly attributed by the authors to inhibition of aldosterone-associated atrial structural and electrical remodeling, to which patients with CKD and type 2 DM are highly predisposed. The importance of this finding is underlined by the results of a stratified Cox proportional hazards model demonstrating an increase in the risk of fatal or nonfatal stroke after a new-onset episode of AF (HR 7.13; 95%CI 4.01–12.70). On the contrary, in pre-specified exploratory analyses of individual components of the composite cardiovascular outcome of the FIDELIO-DKD study, no significant effect on the incidence of nonfatal stroke was observed for finerenone (HR 1.03; 95%CI 0.76–1.38) [86].

The impact of the use of SGLT2is, agents with proven renoprotective effects in patients with CKD [87,88], on the beneficial effects of finerenone observed in the FIDELIO-DKD was recently explored in a pre-specified exploratory sub-analysis of the trial [89]. Overall, 4.4% of patients in the finerenone group and 4.8% in the placebo group were treated with a SGLT2i at baseline. The effect of finerenone compared with placebo on the primary composite renal outcome seemed to be consistent in the subgroups of patients receiving (HR 1.38; 95%CI 0.61–3.10) and those not receiving a SGLT2i (HR 0.82, 95%CI 0.72–0.92) (p-interaction = 0.21). Subgroup analysis for the key secondary renal and cardiovascular outcomes yielded similar results (p-interaction = 0.54 and 0.46, respectively). In addition, numerically fewer episodes of hyperkalemia-related adverse events were observed for patients receiving a SGLT2i at baseline (8.1%) compared to those who did not (18.7%), a phenomenon that was attributed to enhanced potassium excretion due to natriuresis and osmotic diuresis. Overall, these results suggest that MRAs and SGLT2is have complementary actions and that a combined treatment with both agents may provide additional effects above and beyond those associated with each individual agent.

### 3.1.4. Canrenone

To date there are only two studies comparing the effect of canrenone to an active comparator (hydrochlorothiazide) on albuminuria- and kidney-related outcomes in diabetic patients (Table 4). In the study by Fogari et al. [57], among 120 patients with type 2 DM and UACR 60–300 mg/g, UAE decreased by $-45.3\%$ in the canrenone group and by $-20.3\%$ in the hydrochlorothiazide group ($p < 0.01$). No changes in serum levels of creatinine and potassium were noted. In the second study, from the same group [90], authors reported a slight decrease in eGFR with hydrochlorothiazide but not with canrenone, along with a neutral effect on potassium levels.

### 3.1.5. Esaxerenone

Currently, there are results from three studies evaluating the effects of esaxerenone in patients with DKD (Table 4). In the first one, a phase 2b study in 365 patients with type 2 DM with UACR $\geq 45$ to $< 300$ mg/g and eGFR $\geq 30$ mL/min/1.73 m$^2$, esaxerenone showed a significant reduction of albuminuria that was dose-dependent (38% for 1.25 mg, 50% for 2.5 mg and 56% for 5 mg/day) compared with placebo (7%) after 12 weeks of treatment (all $p < 0.001$) [91] (Table 4). In the phase 3 Esaxerenone in Patients with Type 2 Diabetes and Microalbuminuria (ESAX-DN) study, 455 patients with type 2 DM and moderately increased albuminuria were randomized to receive esaxerenone 1.25–2.5 mg/day or placebo on top of an RAS blocker for 52 weeks [92]. Remission of albuminuria was

achieved in 22% of patients in the esaxerenone group compared to 4% in the placebo group (*p* < 0.001), with a mean reduction of UACR by −58% for esaxerenone vs. +8% for placebo at the end of treatment (*p* < 0.001). A significantly higher proportion of patients in the esaxerenone group had a serum potassium of either ≥6.0 mmol/L or ≥5.5 mmol/L at two consecutive occasions compared to placebo (9% vs. 2%, *p* = 0.002), while hyperkalemia led to treatment discontinuation in 4% vs. 1%, respectively. A significantly higher eGFR decline from baseline was noted for esaxerenone than in placebo (−11% vs. −1%). In an open-label 28-week phase 3 study in 56 patients with type 2 DM, UACR ≥ 300 mg/g and eGFR ≥ 30 mL/min/1.73 m$^2$, administration of esaxerenone led to a UACR reduction of −54.6% from baseline and a mean eGFR decline of −8.3 mL/min/1.73 m$^2$; the proportion of patients with serum potassium of ≥6.0 mmol/L or ≥5.5 mmol/L at two occasions was 5.4% [93].

### 3.1.6. Apararenone

The only available data on apararenone originate from a placebo-controlled phase 2 dose-response study with an open-label extension period in 293 Japanese patients with type 2 DM and UACR 50–300 mg/g [62] (Table 4). After 24 weeks of treatment, a significant reduction in UACR ranging between 46.5–62.9% was observed with apararenone, but not with placebo (*p* < 0.001 for all comparisons vs. placebo). The anti-albuminuric effect persisted in the 52-week extension period. A mean increase in serum potassium by 0.2–0.3 mmol/L was evidenced with apararenone. The main pitfall of this study is that only ~65% of participants in each treatment group received an RAS blocker at randomization.

### 3.2. Meta-Analyses of Randomized Clinical Studies with Mineralocorticoid Receptor Antagonists in Diabetic Kidney Disease

In a meta-analysis pooling available results from studies published up to 2019, including also the first two finerenone studies (31 studies, 2767 participants) [14], the addition of an MRA (spironolactone, eplerenone, canrenone, finerenone) alone or on top of an RAS blocker led to a significant reduction of UACR by −24.55%, of 24 h UAE by −32.47% and of urine-protein–creatinine ratio by −53.93%. An eGFR decrease of −2.82 mL/min/1.73 m$^2$ in parallel to an increase in serum potassium by 0.19 mmol/L was noted compared to placebo.

In a recently published meta-analysis of four randomized clinical trials evaluating the efficacy and safety of finerenone in 13,945 patients with DKD [94], the addition of finerenone was associated with a decrease in UACR by −0.30 (95%CI −0.33 to −0.27) and a lower risk of experiencing an eGFR reduction ≥40% (RR 0.85, 95%CI 0.78 to 0.93) compared to placebo. The incidence of hyperkalemia was higher in the finerenone than in the placebo group (RR 2.03, 95%CI 1.83 to 2.26).

**Table 4.** Main studies on the effects of canrenone, esaxerenone and apararenone in albuminuria, renal and cardiovascular outcomes in patients with DKD.

| Study | Study Design | Follow UP | Nr of Patients | Patient Characteristics | Active Treatment | Comparator | Back-Ground Treatment | Main Results |
|---|---|---|---|---|---|---|---|---|
| Fogari et al., 2014 [57] | Open-label, parallel-group RCT | 24 weeks | 120 | T2DM and UACR 60–300 mg/g | Canrenone | Hydrochlorothiazide | Valsartan | ↓ UACR −45.3% for canrenone; −20.3% for hydrochlorothiazide ($p < 0.01$) |
| Derosa et al., 2018 [90] | Double-blind, parallel-group RCT | 12 months | 182 | T2DM and hypertension | Canrenone | Hydrochlorothiazide | ARB | Significant ↓ K only for hydrochlorothiazide ($p < 0.05$); neutral effect for canrenone<br>Significant ↓ eGFR for hydrochlorothiazide ($p < 0.01$)<br>Significant ↑ eGFR for canrenone ($p < 0.05$) |
| Ito et al., 2019 [91] | Double-blind, parallel-group RCT | 12 weeks | 365 | T2DM, UACR 45–300 mg/g, eGFR ≥ 30 mL/min/1.73 m$^2$ | Esaxerenone | Placebo | ACEi or ARB | ↓ UACR −38% for esaxerenone 1.25 mg; −50% for esaxerenone 2.5 mg; −56% for esaxerenone 5 mg; −7% for placebo ($p < 0.001$)<br>Remission of albuminuria: 21% for esaxerenone groups 2.5 and 5.0 mg; 3% for placebo ($p < 0.05$ for both comparisons)<br>HyperK leading to drug discontinuation: 3% for esaxerenone 1.25 and 2.5 mg; 10% esaxerenone 10 mg; 1% for placebo |
| Ito et al., 2020 (ESAX-DN) [92] | Double-blind, parallel-group RCT | 52 weeks | 455 | T2DM, UACR 45–300 mg/g, eGFR ≥ 30 mL/min/1.73 m$^2$ | Esaxerenone | Placebo | ACEi or ARB | ↓ UACR −58% for esaxerenone; +8% for placebo ($p < 0.001$); eGFR −11% for esaxerenone; −1% for placebo<br>Remission of albuminuria: 22% in esaxerenone; 4% in placebo ($p < 0.001$)<br>Time to 1st transition to overt proteinuria: HR 0.23; 95%CI 0.11–0.48 for esaxerenone<br>HyperK episodes (>6.0 mmol/L or ≥5.5 mmol/L at two consecutive occasions): 9% esaxerenone; 2% placebo ($p = 0.002$)<br>HyperK leading to drug discontinuation: 4% for esaxerenone; 1% for placebo |
| Ito et al., 2021 [93] | Open-label, single-arm study | 28 weeks | 56 | T2DM, UACR ≥ 300 mg/g, eGFR ≥ 30 mL/min/1.73 m$^2$ | Esaxerenone | - | ACEi or ARB | ↓ UACR −54.6% ($p < 0.001$); eGFR −8.3 mL/min/1.73 m$^2$ for esaxerenone<br>HyperK episodes (>6.0 mmol/L or ≥5.5 mmol/L at two consecutive occasions): 5.4% for esaxerenone |
| Wada et al., 2021 [62] | Double-blind, parallel-group RCT with open-label extension | 24 weeks and 28 weeks | 293 | T2DM, UACR 50–300 mg/g | Apararenone | Placebo | ACEi or ARB | ↓ UACR at 23 weeks −62.9% apararenone 2.5 mg; −50.8% apararenone 5 mg; −46.5% apararenone 10 mg; +113.7% placebo ($p < 0.001$ vs. placebo for all comparisons)<br>% change in eGFR at 52 weeks: −5.3% (−22.0, +10.5) apararenone 2.5 mg; −10.2% (−34.5, +14.6) apararenone 5 mg; −10.80% (−36.8, +19.1) apararenone 10 mg<br>↑ K at 52 weeks: +0.14 mmol/L (0.006–0.22) apararenone 2.5 mg; +0.18 mmol/L (0.1–0.26) apararenone 5 mg; +0.25 mmol/L (0.16–0.33) apararenone 10 mg |

↓, decrease; ↑, increase; ACEi, angiotensin converting enzyme inhibitor; ARB, angiotensin receptor blocker; CI, confidence interval; eGFR, estimated glomerular filtration rate; hyperK, hyperkalemia; HR, hazard ratio; K, potassium; RCT, randomized controlled trial; T2DM, type 2 diabetes mellitus; UACR, urinary albumin creatinine ratio.

## 4. Future Directions and Perspectives

Overall, the results of the aforementioned randomized controlled provide strong evidence of the beneficial effects of MRAs towards reduction of albuminuria and protection against CKD and CVD progression in patients with DKD. The landscape of treatment of this disease is expected to change in the near future; the use of finerenone will probably be incorporated in forthcoming guidelines and consensus statements of major organizations as first-line treatment for DKD along with SGLT2is to maximize cardio- and nephroprotection. Further controlled trials with cardiovascular and renal outcomes are necessary to shed more light on the presence or absence of similar beneficial effects of the other two nonsteroidal agents, esaxerenone and apararenone.

## 5. Conclusions

Although the use of ACEis/ARBs has been recommended for decades in patients with DKD, and novel, nephroprotective and cardioprotective treatment options such as SGLT2is have been available for some years, important residual risk remains with regards to the rate of CKD and CVD progression. Aldosterone breakthrough represents the main caveat of monotherapy with an ACEi/ARB, leaving the patient exposed to the deleterious genomic and non-genomic effects of aldosterone, including heart and kidney tissue inflammation and fibrosis by MR overactivation. MRA treatment has demonstrated beneficial effects in BP control and reduction of cardiovascular risk in various patient populations. In patients with DKD, first- and second-generation MRAs have been shown to reduce UAE, a strong surrogate marker of renal function decline and progression to ESKD. Despite these promising effects, the absence of hard-outcome data and fear of hyperkalemia has not enabled their extensive use in patients with CKD. Novel third-generation nonsteroidal MRAs, such as finerenone, exhibit greater selectivity than spironolactone and better affinity than eplerenone for the MR, as well as a more balanced distribution between the myocardium and renal tissue in preclinical studies. A pre-specified analysis of a combined population of >13,000 patients with DKD from two major randomized clinical trials of finerenone with hard cardiorenal outcomes demonstrated a significant reduction of −20% in the risk for ESKD and by −22% in the risk for HHF [84]. Overall, nonsteroidal MRAs exhibit an improved benefit–risk ratio than steroidal MRAs, exceeding the risks of combined treatment with an ACEI/ARB. Therefore, the addition of finerenone on top of the standard of care in patients remaining at high risk of progression of their renal and CV disease is expected to confer an additional cardiorenal benefit. Finerenone should receive a clear recommendation in forthcoming guidelines by delaying progression to ESKD and mitigating CV morbidity in patients with DKD.

**Author Contributions:** Conceptualization, P.A.S.; bibliography research, M.-E.A. and M.P.T.; writing—original draft preparation, M.-E.A.; writing—review and editing, M.-E.A., M.P.T. and P.A.S.; supervision, P.A.S. All authors have read and agreed to the published version of the manuscript.

**Funding:** This research received no external funding.

**Institutional Review Board Statement:** Not applicable.

**Informed Consent Statement:** Not applicable.

**Conflicts of Interest:** The authors declare no conflict of interest.

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
