# Peer review of "Role of Mineralocorticoid Receptor Antagonists in Diabetic Kidney Disease"

_kidneydial, doi:10.3390/kidneydial2020019_

Round 1
Reviewer 1 Report
Overall, the manuscript is well written; however, I would suggest you improve the manuscript based on the following suggestions before processing further.
- Add a section describing future directions and perspectives.
- MR and Glucocorticoid receptors (GR) are interrelated to each other. I suggest you describe a few sentences about the roles of glucocorticoid receptors in diabetic nephropathy or kidney disease. Podocyte glucocorticoid receptors and endothelial glucocorticoid receptors?
- What are the new possible drugs or future therapies undergoing development against DKD? These drugs include DPP-4 inhibitor linagliptin, SGLT-2 inhibitor empagliflozin, JAkSTA3 inhibitor, glycolysis inhibitors, ROCK isoforms, SIRT3 and peptide AcSDKP etc. These have been shown promising results in mouse models of diabetic kidney disease. And few of them such as SGLT-2 inhibitors were tested in small- and large-scale clinical trials. What are the superiorities of MR antagonists over others to develop into potential drugs for DKD? Please include this description in the manuscript.
- Describe the Renin-angiotensin-aldosterone System in diabetic kidneys. Include; the diverse effects of RAS inhibitors such as ACEi vs ARB in diabetic kidney disease. ACEi significantly suppressed the kidney injuries whereas ARB did not have such an effect in mouse models of diabetic kidney disease.
- Regulators of MR activators in diabetic kidneys
- How do MR activations cause renal injuries? fibrosis causing mechanisms of MR activators?
- Describe efficacies of different classes of MR antagonists in targeting diabetic kidney disease and discuss their potential side effects
- There are few recent articles available on MR antagonists and Diabetic kidney disease. Cite these articles.
Author Response
Reviewer #1
Overall, the manuscript is well written; however, I would suggest you improve the manuscript based on the following suggestions before processing further.
Comment
1. Add a section describing future directions and perspectives
Author response
We thank Reviewer #1 for this comment. We would like to refer him to our response to the Editor’s comment #1.
Page 18
Comment
2. MR and Glucocorticoid receptors (GR) are interrelated to each other. I suggest you describe a few sentences about the roles of glucocorticoid receptors in diabetic nephropathy or kidney disease. Podocyte glucocorticoid receptors and endothelial glucocorticoid receptors?
Author response
We thank Reviewer #1 for pointing this issue. We would like to refer him to our response to the Editor’s comment #2.
Page 5
Comment
3. What are the new possible drugs or future therapies undergoing development against DKD? These drugs include DPP-4 inhibitor linagliptin, SGLT-2 inhibitor empagliflozin, JAkSTA3 inhibitor, glycolysis inhibitors, ROCK isoforms, SIRT3 and peptide AcSDKP etc. These have been shown promising results in mouse models of diabetic kidney disease. And few of them such as SGLT-2 inhibitors were tested in small- and large-scale clinical trials. What are the superiorities of MR antagonists over others to develop into potential drugs for DKD? Please include this description in the manuscript.
Author response
We thank Reviewer #1 for this comment. However this is an invited review on the “Role of mineralocorticoid receptor antagonists in diabetic kidney disease” for a special issue on DKD. According to the Editor, the use of SGLT-2 inhibitors in DKD will be covered in other articles within this special issue. Some comments on their use are already included in the text: in the following sections: 1) Page 2: “sodium–glucose cotransporter-2 inhibitors (SGLT2i) have shown to offer kidney protection and are now included in recent KDIGO recommendations as preferred 1st line treatment in patients with type 2 diabetes mellitus (DM) and estimated glomerular-filtration-rate (eGFR) ≥30 ml/min/1.73m2 [7].”; 2) now on Page 14 of the revised manuscript: “The impact of use of SGLT2is, agents with proven renoprotective effects… both agents may provide additional effects above and beyond those associated with each individual agent.” according to revision provided for Editor comment #6 and Reviewer #2 comment #4; 3) Page 18: “Although the use of ACEis/ARBs is recommended for decades in patients with DKD, and novel, nephroprotective and cardioprotective treatment options, such as SGLT2is are available for some years”, so we have not proceeded to any further revisions. We believe that reference to DPP-4 inhibitors, JAkSTA3 inhibitors, glycolysis inhibitors, and other peptides is similarly outside of the scope of this invited review.
Page 2, 14, 18
Comment
4. Describe the Renin-angiotensin-aldosterone System in diabetic kidneys. Include; the diverse effects of RAS inhibitors such as ACEi vs ARB in diabetic kidney disease. ACEi significantly suppressed the kidney injuries whereas ARB did not have such an effect in mouse models of diabetic kidney disease.
Author response
We thank Reviewer #1 for this comment. However this is an invited review on the “Role of mineralocorticoid receptor antagonists in diabetic kidney disease” for a special issue on DKD. According to the Editor, the use of renin-angiotensin-aldosterone system inhibitors in DKD, will be covered in other articles dedicated to this subject within this special issue, so we have not proceeded to any further revision, as these agents are already mentioned in Page 1 of the initially submitted manuscript: “Over the last decades, the use of renin–angiotensin-system (RAS) blockers has been the mainstay for retarding progression of DKD [5,6], along with lifestyle modifications, blood pressure (BP) and glycemic control [7].”
Page 1
Comment
5. Regulators of MR activators in diabetic kidneys
Author response
We thank Reviewer #1 for this comment. We would like to refer him to Page 4 of the Manuscript where we already mention the emergence of the crucial role of regulators of MR activation: “In addition to the above, over the past years the crucial role of coregulatory molecules in mediating the genomic response to nuclear receptor activation has emerged [36]. Accumulated evidence suggests that upon ligand binding, the transcription of effector proteins is modulated via recruitment of coactivator or corepressor proteins according to distinct MR conformations induced by binding of different agonist ligands [25,36].” We believe this is an adequate reference on this issue, as a more detailed analysis on this section would increase the length of the manuscript outweighing the importance of pathophysiology over the total amount of evidence that needs to be presented in this invited review on the “Role of mineralocorticoid receptor antagonists in diabetic kidney disease”, where clinical effects need to be emphasized.
Page 4
Comment
6. How do MR activations cause renal injuries? fibrosis causing mechanisms of MR activators?
Author response
We thank Reviewer # 1 for this comment and we would like to refer him to our response to the Editor’s comment #3.
Page 4
Comment
7. How do MR activations cause renal injuries? fibrosis causing mechanisms of MR 7. Describe efficacies of different classes of MR antagonists in targeting diabetic kidney disease and discuss their potential side effects
Author response
We thank Reviewer #1 for this comment. In our manuscript, we believe we described in much extend all available to date data on the efficacy as well as on safety profile of each class and type of agent is more than extensively discussed in dedicated sections: “3.1.1 Spironolactone”, “3.1.2 Eplerenone”, “3.1.3 Finerenone”, “3.1.4 Canrenone”, ‘3.1.5 Esaxerenone”, “3.1.6 Apararenone”, Moreover, we present extensive information in Tables 1, 2, 3, and 4 where all available results were presented in a concise but comprehensive way.
Pages 5-17, Tables 1,2,3,4
Comment
8. There are few recent articles available on MR antagonists and Diabetic kidney disease. Cite these articles.
Author response
We thank Reviewer #1 for this comment. We believe that we had covered all available literature from original articles at the time of Manuscript’s submission (February 2022), including even results from a meta-analysis that was accepted in January 2022 (Zheng Y et al. Kidney Blood Press Res. 2022).
Reviewer 2 Report
The authors reported the review of the role of mineralocorticoid receptor antagonists in DKD. This article was well written overall.
The authors should add more information about SGLT2i in DKD in Introduction section. SGLT2i is getting a lot of attention in DKD and CKD.
The author showed the FIGARO-DKD study. In line 329-344, they reported that Finerenone was associated with a lower risk for the key secondary renal outcome, a composite of kidney events. Is it true? I think that Finerenone was not associated with a lower risk for the key secondary renal outcome, a composite of kidney events.
FIDELIO-DKD study was only study which showed the effect of MRA to a composite of kidney events as primary endpoint. The authors should emphasize this point. All other studies only showed an effect on albuminuria, which is a surrogate marker of kidney dysfunction.
Moreover, only 4.4% patients in FIDELIO-DKD study have SGLT2 inhibitors. The results of the study might differ if more patients have SGLT2 inhibitors, which have strong renoprotective effects. I recommend that the authors explain this point.
Author Response
Reviewer #2
The authors reported the review of the role of mineralocorticoid receptor antagonists in DKD. This article was well written overall
Comment
1. The authors should add more information about SGLT2i in DKD in Introduction section. SGLT2i is getting a lot of attention in DKD and CKD.
Author response
We thank Reviewer #2 for this comment. However this is an invited review on the “Role of mineralocorticoid receptor antagonists in diabetic kidney disease” for a special issue on DKD. According to the Editor, the use of SGLT-2 inhibitors in DKD, will be covered in other articles within this special issue, so we have not proceeded to any further revision, as these agents are mentioned in the following sections: 1) Page 2: “sodium–glucose cotransporter-2 inhibitors (SGLT2i) have shown to offer kidney protection and are now included in recent KDIGO recommendations as preferred 1st line treatment in patients with type 2 diabetes mellitus (DM) and estimated glomerular-filtration-rate (eGFR) ≥30 ml/min/1.73m2 [7].”; 2) now on Page 14: “ The impact of use of SGLT2is, agents with proven renoprotective effects … both agents may provide additional effects above and beyond those associated with each individual agent.” according to revision provided for Reviewer #2 comment #4; 3) Page 18: “Although the use of ACEis/ARBs is recommended for decades in patients with DKD, and novel, nephroprotective and cardioprotective treatment options, such as SGLT2is are available for some years”
Pages 2, 14, 18
Comment
2. The author showed the FIGARO-DKD study. In line 329-344, they reported that Finerenone was associated with a lower risk for the key secondary renal outcome, a composite of kidney events. Is it true? I think that Finerenone was not associated with a lower risk for the key secondary renal outcome, a composite of kidney events.
Author response
We thank the Editor for this comment. At this point we would like to cite for the Editor the relevant part of our initially submitted manuscript to which the Reviewer #1 is referring. We state that “Finerenone was also associated with a marginally lower risk for the key secondary renal outcome, a composite of kidney failure, sustained eGFR decrease of ≥40% from baseline, or renal death (HR 0.87; 95%CI 0.76-1.01). With regards to different components of the secondary outcome, significantly lower rates of progression to ESKD were observed with finerenone (0.9%) than with placebo (1.3%) (HR 0.64; 95%CI 0.41-0.995).” As commented by the investigators of the FIGARO-DKD study, significance was not achieved here (Pitt B, NEJM 2021), but given the CI (0.76-1.01) of the composite secondary renal outcome, they were marginally reaching statistical significance so the term marginally was used. This particular effect is clearly significant when the “harder” end-point of “sustained eGFR decrease of ≥57% from baseline” was included in the composite, a fact supporting the probable unsuitability of eGFR decrease of ≥40% to describe true progression of CKD (Pitt B, NEJM 2021). To further clear up this issue, we have now added the following phrase in the end of page 14 of the Revised Manuscript, now reading: “a marginally but not statistically significant lower risk…”
Page 14
Page 14
Comment
3. FIDELIO-DKD study was only study which showed the effect of MRA to a composite of kidney events as primary endpoint. The authors should emphasize this point. All other studies only showed an effect on albuminuria, which is a surrogate marker of kidney dysfunction.
Author response
We thank you for this comment. Actually this was the reason for initially including the 2nd paragraph of Section 3.1.3. Finerenone in Page 11 of the Manuscript, where we state that: “The clinical trials discussed above were focused in changes of albuminuria and were often not adequately powered to detect changes in the slope of eGFR or progression of CKD and CVD. Despite the fact that changes in albuminuria have proven to be strongly associated with lower hazard for renal clinical endpoints (ESKD, fall of eGFR <15ml/min/1.73m2) and lower risk of death [82,83], these represent an intermediate marker of kidney disease. Thus, from an efficacy point of view, the only currently available data on the impact of MRAs on long-term cardiorenal protection originate from the two recently published large-scale phase 3 randomized clinical trials of finerenone, the FIDELIO-DKD (Finerenone-in-reducing-kidney-failure-and-disease-progression-in- Diabetic-Kidney-Disease) [23] and the FIGARO-DKD (Finere-none-in-reducing-cardiovascular-mortality-and-morbidity-in-Diabetic-Kidney-Disease) [24].” In order to underline again emphasis placed on that point, we have repeated this phrase when presenting results of the FIDELIO DKD study (Page 14 of the revised manuscript), now reading: “It should be further emphasized that these results represent the first available evidence on beneficial effects of an MRA agent on hard renal and cardiovascular outcomes.”
Page 14
Comment
4. Moreover, only 4.4% patients in FIDELIO-DKD study have SGLT2 inhibitors. The results of the study might differ if more patients have SGLT2 inhibitors, which have strong renoprotective effects. I recommend that the authors explain this point.
Author response
We thank the Editor for pointing this important issue. According to Reviewer’s #2 suggestion, we have now included in the end of Section “3.1.3. Finerenone” in Page 14 of the Revised Manuscript a relative comment on the impact of SGLT-2i on the renoprotective effects of finerenone based on the results of a secondary analysis published by Rossing et al. The text now reads: “The impact of use of SGLT-2is, agents with proven renoprotective effects in patients with CKD [87,88], on the beneficial effects of finerenone observed in the FIDELIO-DKD was recently explored in a prespecified exploratory subanalysis of the trial [89]. Overall 4.4% of patients in the finerenone group and 4.8% in the placebo group were treated with an SGLT-2i at baseline. The effect of finerenone compared with placebo on the primary composite renal outcome seemed to be consistent in the subgroups of patients receiving (HR 1.38; 95%CI 0.61–3.10) and those not receiving an SGLT-2i (HR 0.82, 95%CI 0.72–0.92) (p-interaction=0.21). Subgroup analysis for the key secondary renal and cardiovascular outcomes yielded similar results (p-interaction=0.54 and 0.46 respectively). In addition, numerically fewer episodes of hyperkalemia-related adverse events were observed for patients receiving an SGLT-2i at baseline (8.1%) compared to those who did not (18.7%), a phenomenon that was attributed to enhanced potassium excretion due to natriuresis and osmotic diuresis. Overall these results suggest that MRAs and SGLT-2i have complementary actions and that a combined treatment with both agents may provide additional effects above and beyond those associated with each individual agent.”
Pages 14, 17
Pages 14, 17
Round 2
Reviewer 1 Report
The authors have addressed well to the comments.
Thanks